# SceneMaker: Open-set 3D Scene Generation with Decoupled De-occlusion and Pose Estimation Model

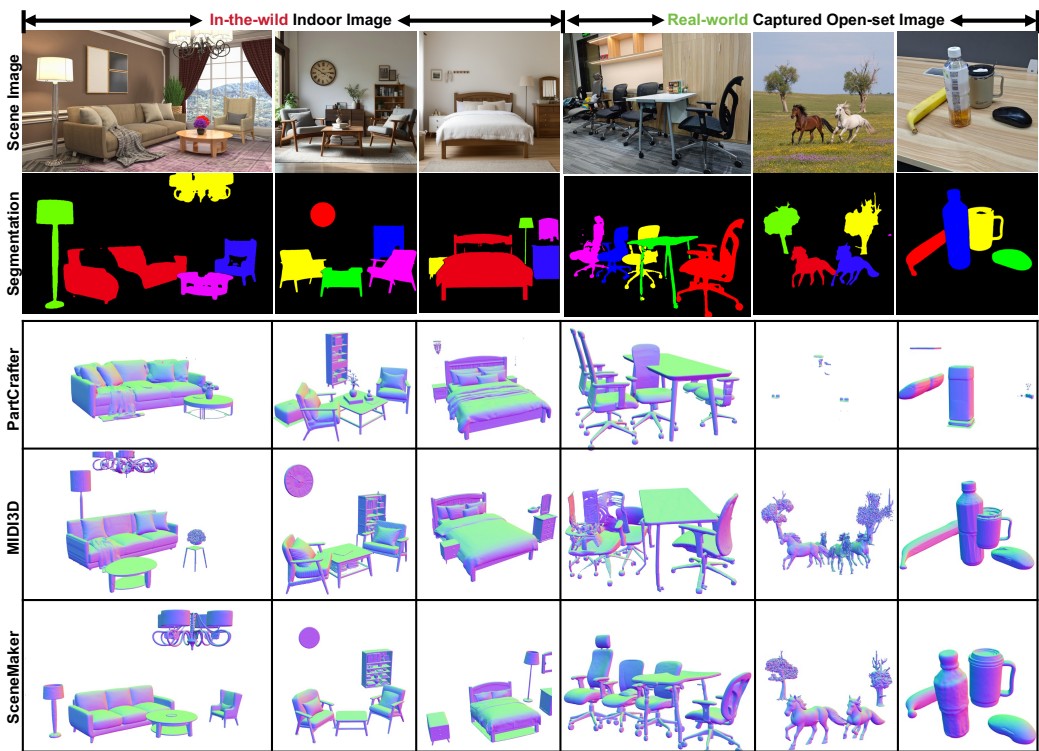

Figure 1: Our method achieves better performance on both **in-the-wild** indoor scenes and **real-world captured** open-set images. Existing methods lack sufficient de-occlusion and open-set priors.

## Abstract

We propose a decoupled 3D scene generation framework called **SceneMaker** in this work. Due to the lack of sufficient open-set de-occlusion and pose estimation priors, existing methods struggle to simultaneously produce high-quality geometry and accurate poses under severe occlusion and open-set settings. To address these issues, we first decouple the de-occlusion model from 3D object generation, and enhance it by leveraging image datasets and collected de-occlusion datasets for much more diverse open-set occlusion patterns. Then, we propose a unified pose estimation model that integrates global and local mechanisms for both self-attention and cross-attention to improve accuracy. Besides, we construct an open-set 3D scene dataset to further extend the generalization of the pose estimation model. Comprehensive experiments demonstrate the superiority of our decoupled framework on both indoor and open-set scenes.

## 1 INTRODUCTION

Open-set 3D scene generation aims to synthesize 3D scenes containing arbitrary objects in any open-world domain from a single image. It is a fundamental task with high demand in AIGC and embodied AI, including applications such as 3D asset creation, simulation environment construction, and 3D perception for decision-making. However, limited scene datasets (Fu et al., 2021; Dai et al., 2017; Azinović et al., 2022) have confined most existing methods (Tang et al., 2024; Dahnert et al., 2024; Liu et al., 2022; Dai et al., 2024) to constrained domains like indoor scenes.

Recently, the advent of large-scale 3D object datasets (Deitke et al., 2023) has driven rapid progress in open-set 3D object generation models (Zhang et al., 2024b; Wu et al., 2024; Li et al., 2024; 2025b; Xiang et al., 2025; Zhao et al., 2025; Li et al., 2025a), and emerging methods (Yao et al., 2025; Huang et al., 2024; Lin et al., 2025; Meng et al., 2025; Dogaru et al., 2024) are beginning to extend scene generation toward open-set settings. Despite all the progress, existing methods still struggle to simultaneously produce high-quality geometry and accurate poses under severe occlusion and open-set settings as shown in Figure 1.

The root cause is the model's insufficient open-set priors for de-occlusion and pose estimation. As illustrated in Figure 2, a 3D scene generation model requires three key open-set priors in columns: de-occlusion, object geometry, and pose estimation. The availability of these priors varies across scene, object, and image datasets in rows (Fu et al., 2021; Azinović et al., 2022; Deitke et al., 2023; Schuhmann et al., 2022; Deng et al., 2009). Paths in different colors represent various scene generation methods with different prior sources. Existing scene-native methods (**yellow path**) (Tang et al., 2024; Dahnert et al., 2024; Liu et al., 2022; Dai et al., 2024) attempt to learn all the three priors exclusively from scene datasets, where the availability of open-set priors is limited. Object-native methods (**green path**) (Yao et al., 2025; Huang et al., 2024; Lin et al., 2025; Meng et al., 2025; Dogaru et al., 2024; Qu et al., 2025; Wu et al., 2025) further leverage large-scale 3D object datasets to learn sufficient open-set object geometry priors. However, the open-set priors for de-occlusion and pose estimation still remain insufficient due to the limited datasets, leaving these challenges unresolved. Meanwhile, existing pose estimation methods (Wen et al., 2024; Zhang et al., 2023; 2024a) suffer from performance degradation in scene generation task, primarily due to missing size prediction and the absence of tailored attention mechanisms for different pose variables.

In this paper, we further advance 3D scene generation towards open-set scenarios by addressing the critical issue of insufficient de-occlusion and pose estimation priors, as shown in Figure 2 (**red path**). Specifically, we first construct a decoupled framework that divides 3D scene generation into three distinct tasks based on the necessary priors: de-occlusion, 3D object generation, and pose estimation. Each task is trained separately on image datasets, 3D object datasets, and scene datasets, respectively. The decoupled framework ensures that each task can maximize the learning of its corresponding open-set priors, preventing quality degradation caused by the cross-impact of data on tasks, such as geometry collapse of small objects and pose shifting resulting from the joint representation of geometry and pose in Figure. 1.

Second, we develop a robust de-occlusion model by leveraging image datasets for open-set occlusion prior. Image datasets are significantly larger than 3D datasets, encompassing a broader range of open-set objects and exhibiting more diverse occlusion patterns. To maintain the sufficient open-set priors, we adopt the image editing model (Labs et al., 2025) as the initialization. Then, we finetune it on our 10K image de-occlusion dataset with three carefully designed occlusion patterns to further enhance its de-occlusion capability, resulting in the final de-occlusion model. Compared with existing 3D object-based methods (Wu et al., 2025; Huang et al., 2024), our model achieves higher quality and more text-controllable results under severe occlusion and open-set conditions.

Third, we propose a unified pose estimation model along with a 200K scene dataset for better performance and open-set generalization. Since 3D object generation models (Zhang et al., 2024b; Zhao et al., 2025; Li et al., 2024; Chen et al., 2025) usually output normalized objects in canonical space for better geometry, existing methods (Wen et al., 2024; Zhang et al., 2023; 2024a) often miss size prediction when they are employed in scene generation task. Thus, we propose a unified diffusion-based pose estimation model, which directly predicts object rotation, translation, and size conditioned on point clouds, images, and object geometry. Compared to existing methods (Yao et al., 2025), we introduce both single object and multi-object self-attention mechanism to ensure interactions between objects for coherent relationships. Moreover, we design a decoupled cross-attention mechanism, where rotation attends to canonical object conditions, while translation and scale attend

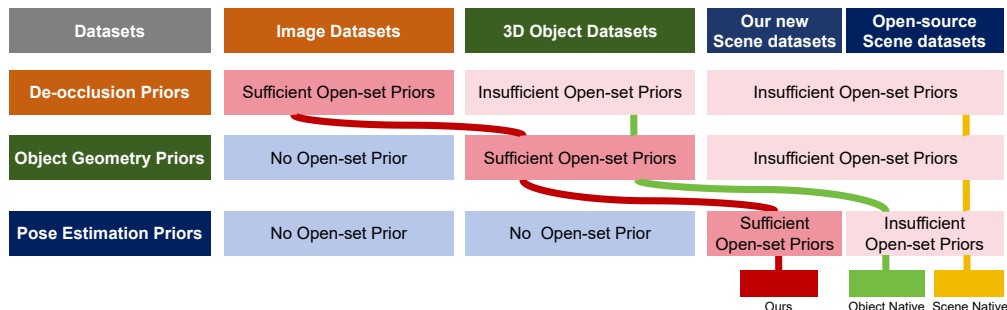

Figure 2: **The analysis of prior sources in different methods.** The table shows that the availability of required open-set priors (column) varies across different datasets (row). Paths in different colors represent various scene generation methods with different prior sources. Existing methods (**yellow path** and **green path**) lack sufficient open-set priors for de-occlusion and pose estimation due to the limited datasets. We further leverage image datasets for de-occlusion and collect new scene datasets for pose estiamtion to achieve better open-set performance(**red path**).

to scene-level conditions, to further improve accuracy. Additionally, to extend open-set capability, we construct a large-scale synthetic dataset of 200K scenes using Objaverse (Deitke et al., 2023) objects and mix it with existing scene datasets during training.

Finally, comprehensive experiments demonstrate that our model achieves state-of-the-art performance in both object geometry quality and pose accuracy on both indoor and open-set test sets. As our model is inherently compatible with other image inputs such as videos and multi-images, we further discuss its potential upper bound across these modalities.

In summary, our contributions are as threefold:

- We construct a decoupled 3D scene generation framework called **SceneMaker** that fully exploits existing datasets to learn sufficient open-set priors for de-occlusion and pose estimation, achieving superior performance in comprehensive experiments.
- We develop a robust de-occlusion model by leveraging image datasets for open-set occlusion priors and enhancing it with our 10K object image de-occlusion dataset.
- We propose a unified pose estimation diffusion model that directly predicts each object's 6D pose and size, introducing both local and global attention mechanisms to enhance accuracy. And we further curate a 200K synthesized scene dataset for open-set generalization.

## 2 RELATED WORK

### 2.1 3D SCENE GENERATION

3D scene generation is in high demand for AIGC and embodied AI, serving as a foundation task for real-to-sim applications. Based on the source of 3D objects, existing methods fall into two categories: generation-based and retrieval-based. Retrieval-based methods (Dai et al., 2024) retrieve 3D objects from offline libraries but struggle to generalize to open-set scenarios due to limited asset diversity. Generation-based methods directly generate 3D objects from images and can be categorized into scene-native and object-native methods. Scene-native methods (Tang et al., 2024; Dahnert et al., 2024; Liu et al., 2022) directly learn from scene datasets (Fu et al., 2021; Azinović et al., 2022; Dai et al., 2017) but are limited to specific domains like indoor scenes. Object-native methods further leverage open-set 3D object datasets (Deitke et al., 2023) to improve object geometry quality. A series of methods (Yao et al., 2025; Huang et al., 2024; Lin et al., 2025; Meng et al., 2025; Dogaru et al., 2024) directly generate object geometry in the scene space. However, due to the limitations of scene datasets and the coupled representation, they often suffer from obvious degradation on images with severe occlusion or small objects. Another series of methods (Yao et al., 2025) decouple geometry generation and pose estimation to improve the open-set performance. But they lack scene-level interactions during pose estimation, leading to inaccurate relative poses. Fundamentally, existing methods lack sufficient de-occlusion and pose estimation priors. We supplement both open-set priors by leveraging image datasets for de-occlusion and proposing a unified model along with synthetic scene datasets for pose estimation.

### 2.2 OBJECT GENERATION UNDER OCCLUSION

With the emergence of large-scale open-set 3D object datasets (Deitke et al., 2023), a number of native 3D object generation works (Zhang et al., 2024b; Wu et al., 2024; Li et al., 2024; 2025b; Xiang

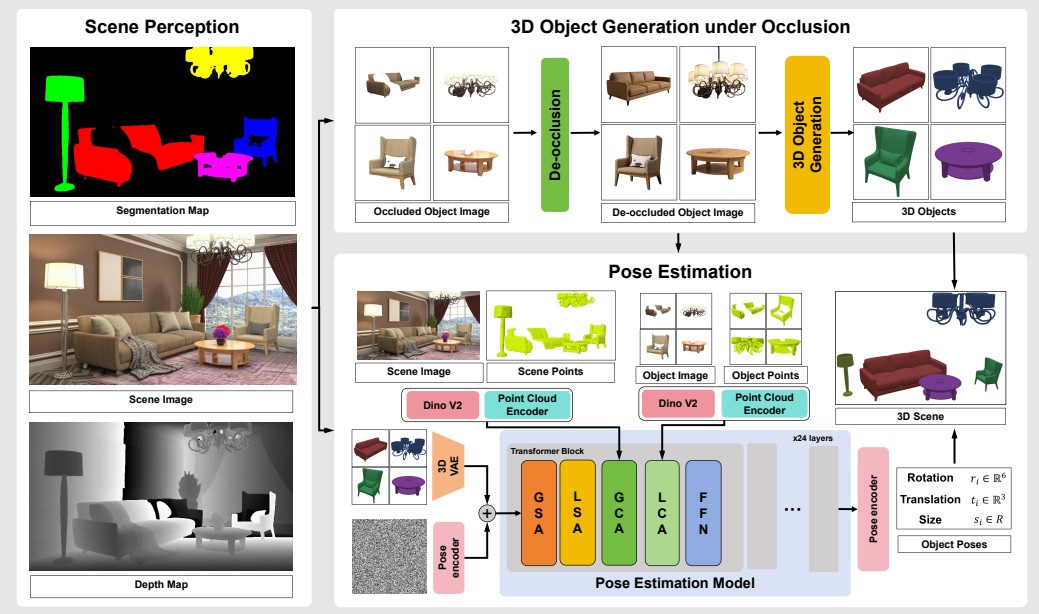

Figure 3: **The Framework of SceneMaker.** Our framework consists of scene perception, 3D object generation under occlusion, and pose estimation. We decouple the de-occlusion model from 3D object generation. We construct a unified pose estimation model that incorporates both global and local attention mechanisms. GSA, LSA, GCA, LCA, and FFN denote global self-attention, local self-attention, global cross-attention, local cross-attention, and feed-forward network, respectively.

et al., 2025; Zhao et al., 2025) have achieved impressive results. However, generating 3D objects under occlusion conditions is more aligned with the needs of scene generation and still requires further exploration. Most existing methods (Chu et al., 2023; Zhou et al., 2021; Stutz & Geiger, 2018; Cui et al., 2024) model the task as 3D completion, where partial geometry is derived from images and subsequently completed using 3D generation models. Recently, some methods (Wu et al., 2025; Cho et al., 2025) additionally use occluded images and masks as supplementary information to achieve better performance. Since 3D generation models already possess sufficient geometric priors, the bottleneck is the lack of de-occlusion priors. Image datasets, which contain more diverse occlusion patterns than 3D datasets, have not been fully utilized. We address this by decoupling the de-occlusion model and leveraging image datasets for training to enhance quality and controllability.

### 2.3 POSE ESTIMATION

Model-based pose estimation aims to predict poses based on the given CAD model. Existing methods (Zheng et al., 2023; Tian et al., 2020; Wang et al., 2019; Zhang et al., 2022) have achieved impressive performance on predefined classes. Recent works (Shugurov et al., 2022; Labbé et al., 2022; Wen et al., 2024; Zhang et al., 2023; 2024a) further extend the task to arbitrary objects with regression or diffusion models. However, they lack the size prediction when they are employed on scene generation task. CAST3D (Yao et al., 2025) address the issue with a point diffusion model, but it lacks both interaction between objects and decoupled mechanism with conditions from different spaces. We propose a unified pose estimation diffusion model with both local and global attention mechanisms to improve accuracy.

## 3 METHOD

In this work, we construct a decoupled 3D scene generation framework called **SceneMaker** that fully exploits existing datasets to learn sufficient open-set priors. In Section 3.1, we formulate and overview the whole scene generation framework. In Section 3.2 we introduce how to leverage image datasets for decoupled de-occlusion model in 3D object generation. In Section 3.3, we propose the unified pose estimation model and extend open-set generalization with synthetic datasets.

### 3.1 FRAMEWORK

As shown in Figure 3, given a single scene image $X$ containing multiple objects $X = \{x_1, x_2, ..., x_n\}$, our scene generation framework aims to generate a consistent 3D scene $Z$ con-

taining corresponding 3D objects $Z = \{z_1, z_2, ..., z_n\}$. Our framework consists of three modules: scene perception, 3D object generation under occlusion, and pose estimation, which are formally following the subsequent automated steps.

1) Utilize Grounded-SAM (Ren et al., 2024) to segment object masks $M = \{m_1, m_2, ..., m_n\}$. Apply the mask on the scene image $X$ to obtain occluded object images $I = \{i_1, i_2, ..., i_n\}$.

2) Utilize MoGe (Wang et al., 2025b) to estimate scene depth map $D$. Apply mask $M$ on the depth and project pixels into 3D space to obtain point clouds $C = \{c_1, c_2, ..., c_n\}$.

3) Acquire de-occluded object images $I^d = \{i_1^d, i_2^d, ..., i_n^d\}$ with $\epsilon_\theta^d(I_t^d; t, I) \rightarrow I^d$, where $\epsilon_\theta^d$ denotes our decoupled de-occlusion model and $t$ denotes timesteps in diffusion models.

4) Generate 3D object geometry $O = \{o_1, o_2, ..., o_n\}$ based on de-occluded images $I^d$ with $\epsilon_\theta^o(O_t; t, I^d) \rightarrow O$, where $\epsilon_\theta^o$ denotes the 3D generation model.

5) Estimate object poses $P = \{p_1, p_2, ..., p_n\}$ based on point clouds, images and object geometry with $\epsilon_\theta^p(P_t; t, X, M, I, C, O) \rightarrow P$, where $\epsilon_\theta^p$ denotes the pose estimation model. Here the object poses contain rotation, translation, and size: $p_i = \{r_i, t_i, s_i\}$.

6) Composite generated object geometry and estimated poses into the final scene: $Z = \{O, P\}$.

In this formulation, we construct the decoupled 3D scene generation framework that fully exploits existing datasets to learn sufficient open-set priors.

## 3.2 Object Generation with Decoupled De-occlusion Model

After obtaining the depth map and segmentation masks from the scene perception module, we aim to generate 3D objects with the high-quality geometry based on occluded object images. However, existing methods often struggle to generate high-quality geometry under severe occlusion. The main challenge is that models lack sufficient open-set occlusion priors due to limited 3D datasets.

Image datasets are significantly larger than 3D datasets, encompassing a broader range of open-set objects and exhibiting more diverse occlusion patterns. Therefore, compared with existing methods, we further decoupled the de-occlusion model and train it on image datasets for richer occlusion priors. The de-occlusion model is formulated as follow:

$$\epsilon_\theta^d(I_t^d; t, I) \rightarrow I^d, \tag{1}$$

where $\epsilon_\theta^d$, $I$, $I^d$, $t$ denote our decoupled de-occlusion model, occluded images, de-occluded images, and timesteps in diffusion models, respectively.

Since existing 3D native object generation models (Zhao et al., 2025; Zhang et al., 2024b; Li et al., 2024; Xiang et al., 2025) have achieved impressive performance, we simply adopt existing methods for image-3d generation after de-occlusion, as shown in Equation 2:

$$\epsilon_\theta^o(O_t; t, I^d) \rightarrow O, \tag{2}$$

where $\epsilon_\theta^o$ and $O$ denote the 3D generation model and generated 3D objects, respectively.

### 3.2.1 De-occlusion Model

To acquire sufficient open-set priors, we directly use the image editing model (Labs et al., 2025) as the initialization for the de-occlusion model. Although both editing (Labs et al., 2025) and inpainting (Ju et al., 2024) models can achieve de-occlusion, their performance is often suboptimal in cases of severe occlusion. The fundamental cause is the lack of diverse and severe occlusion patterns in the training data. To address this, we construct an additional 10K object image de-occlusion dataset to finetune the model and enhance its de-occlusion capability.

**De-occlusion Datasets.** We first use GPT (Achiam et al., 2023) to generate detailed captions of objects, and then employ an image generation model (Flux, 2024) to produce high-quality target images. Considering that occluded images are derived from the segmentation model (Ren et al., 2024) based on predefined class labels (Liu et al., 2024), we generate 20 captions per class, and further expand them as detailed as possible to ensure high-quality images. Meanwhile, we create a universal template as the de-occlusion text prompt for all classes. Next, we carefully design three masking strategies to simulate real-world occlusions: object cutouts without background for object occlusion, right-angle cropping for image borders, and random brush strokes for user prompts. Finally, the final de-occlusion dataset is constructed by 10K triplets formed by masked images, de-occlusion text prompts, and target images.

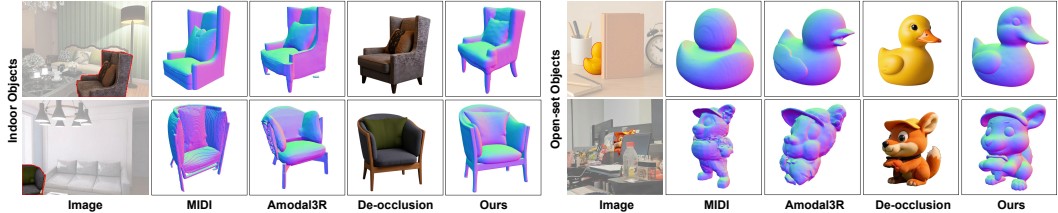

Figure 4: Qualitative comparison of de-occlusion models. Our model has better performance on both indoor and open-set objects, especiallt under severe occlusion.

Figure 5: Qualitative comparison of object generation under occlusion. Our model has better performance on both indoor and open-set objects.

### 3.2.2 COMPARISON

**De-occlusion.** We conduct both quantitative and qualitative experiments to demonstrate the superiority of our de-occlusion model. We mainly compare our model with the state-of-the-art methods in image painting (Ju et al., 2024) and image editing (Labs et al., 2025). We

| Methods | PSNR ↑ | SSIM ↑ | CLIP ↑ |
|---|---|---|---|
| BrushNet | 11.07 | 0.6760 | 0.2659 |
| Flux Kontext | 13.91 | 0.7309 | 0.2674 |
| Ours | **15.03** | **0.7566** | **0.2698** |

Table 1: Quantitative comparison of de-occlusion.

evaluate these methods on our collected validation set of 1K images spanning over 500 classes. We use PSNR and SSIM between the prediction and ground truth images, as well as the CLIP score (Radford et al., 2021) between the prediction image and class labels, as evaluation metrics. As shown in Figure 4 and Table 1, our de-occlusion model achieve better performance on both indoor and open-set scenes, especially under severe occlusions.

**Object Generation under Occlusion.** To demonstrate the superiority of our decoupled pipeline, we compare it with existing 3D native methods (Wu et al., 2025; Huang et al., 2024) on the 3D object generation task. As shown in Table 2, we conduct quantitative experiments on 3D Front datasets (Fu et al., 2021) and im-

| Methods | CD ↓ | F-Score ↑ | Volume IoU ↑ |
|---|---|---|---|
| MIDI | 0.0508 | 0.5533 | 0.4214 |
| Amodal3R | 0.0443 | 0.7124 | 0.5279 |
| Ours | **0.0409** | **0.7454** | **0.5985** |

Table 2: Quantitative comparison of object generation under occlusion.

ages with more severe occlusions rendered by InstPifu (Liu et al., 2022). We further conduct qualitative experiments on indoor and open-set scenes in Figure 5. Both qualitative and quantitative results show that our decoupled framework achieves superior performance in object generation under occlusion across both indoor and open-set scenes.

### 3.3 UNIFIED POSE ESTIMATION MODEL

The goal of the pose estimation model is to predict each object's rotation $R$, translation $T$, and size $S$ in the scene based on its canonical geometry $O$. Existing methods (Wen et al., 2024; Zhang et al., 2024a; 2023; Huang et al., 2024; Yao et al., 2025) mainly face three challenges. First, they often miss size prediction when they are employed in scene generation task, since object geometries are usually generated in canonical space. Second, they do not properly decouple different pose variables when interacting with scene-level and object-level features, resulting in performance degradation. To address these two issues, we propose a unified pose estimation model that incorporates both global and local attention mechanisms in Section 3.3.1. Third, existing methods often struggle on open-set scenarios due to limited datasets. We build a large-scale open-set dataset containing over 200K synthesized scenes to tackle the generalization challenge in Section 3.3.2.

Figure 6: **Attention mechanisms in the pose estimation model.** The global self-attention module enables tokens of all objects in the scene to interact with each other. The local cross-attention module enables rotation tokens independently interact with conditions in the object canonical space. The global cross-attention module enables translation and size tokens attend to scene-level conditions.

### 3.3.1 PIPELINE

As shown in Figure 3, we propose a unified pose estimation model that introduces both global and local attention mechanisms specific for the scene generation task. We directly incorporate object size into the prediction and jointly estimate it with rotation and translation, to address the adaptation challenge in scene generation task. Specifically, we take scene images $X$, scene masks $M$, cropped object images $I$, point clouds $C$, and object geometries $O$ as inputs, and predict object rotation $R$, translation $T$, and size $S$ as outputs, where rotation is represented in 6D.

To improve learning efficiency, all scenes are normalized to a unified space for pose estimation. Since all pose variables can be well represented within a Gaussian distribution, we employ the diffusion model (Ho et al., 2020; Lipman et al., 2022; Peebles & Xie, 2023) for pose estimation from a generative perspective, where poses are denoised from Gaussian noise with the input modalities serving as conditioning signals. The final formulation can be represented in Equation 3.

$$\epsilon_\theta^p(P_t; t, X, M, I, C, O) \to P,$$
$$P = \{R, T, S\}, \tag{3}$$

where $\epsilon_\theta^p$, $t$ denote the pose estimation model and timestep in diffusion models, respectively.

As shown in Figure 3, the trainable object pose encoder and decoder are composed of MLPs. Object geometries, images, and point clouds are encoded into features using a pretrained 3D object VAE, Dinov2 (Oquab et al., 2023), and a point encoder pretrained on 3D reconstruction tasks, all of which are kept frozen during training. Object geometry is injected through concatenation with pose tokens, while image and point cloud features are injected via cross-attention. We implement our model using a flow matching framework (Lipman et al., 2022) with a DiT architecture (Peebles & Xie, 2023), where each transformer block consists of global and local self-attention, global and local cross-attention, and a feed-forward network.

**Attention Mechanisms.** As shown in Figure 6, we adopt both global and local mechanisms for self-attention and cross-attention. Each pose variable is separately encoded as a token, so each object in the diffusion model is uniquely represented by a quadruple of tokens: rotation, translation, size, and geometry. The local self-attention module enables the interaction inside the quadruple of each object. The global self-attention module enables tokens of all objects in the scene to interact with each other, leading to more coherent relative object poses. Considering that rotation can be independently estimated in the object canonical space and scene-level conditions provide little benefit, we introduce a local cross-attention module, allowing the rotation token to attend only to the cropped object image and normalized object point cloud. Meanwhile, we retain a global cross-attention module for the translation and size tokens, allowing them to attend to the scene-level point cloud and image. This decoupled attention mechanism is demonstrated to improve model performance in our comprehensive experiments.

### 3.3.2 OPEN-SET SCENE DATASETS

Since existing datasets currently lack the necessary prior for training a 3D scene generation model in an open-set domain, we addressed this by constructing our own training data. This involved using a carefully curated subset of the existing Objaverse (Deitke et al., 2023) dataset along with Blender (ble, 2025). A significant number of models in Objaverse are either scanned data or have low-quality textures and materials, which necessitated a rigorous curation process. To filter the models, we assessed their material information, excluding any that were transparent, lacked a BSDF node, or did not have an albedo map. To further refine the selection, we also excluded models with pure or excessively dark albedo colors. Ultimately, this process resulted in a high-quality subset of 90k models with a superior appearance to construct a dataset of 200k scenes for our work.

| Method | 3D-Front | | | | | Open-set | | | | |
|---|---|---|---|---|---|---|---|---|---|---|
| | CD-S↓ | F-Score-S↑ | CD-O↓ | F-Score-O↑ | IoU-B↑ | CD-S↓ | F-Score-S↑ | CD-O↓ | F-Score-O↑ | IoU-B↑ |
| PartCrafter | 0.1846 | 0.3844 | - | - | - | 0.2171 | 0.2613 | - | - | - |
| MIDI3D | 0.1672 | 0.3420 | 0.0663 | 0.5495 | 0.3855 | 0.1425 | 0.3211 | 0.0807 | 0.5602 | 0.5079 |
| SceneMaker (w/o open-set data) | **0.0381** | **0.6840** | **0.0681** | 0.6160 | 0.7658 | 0.1538 | 0.4644 | 0.0847 | 0.5771 | 0.6248 |
| SceneMaker(Ours) | 0.0470 | 0.6312 | 0.0885 | **0.6812** | **0.7693** | **0.0285** | **0.6125** | **0.0671** | **0.5948** | **0.7549** |

Table 3: **Quantitative comparison with scene generation methods.**

We composed each scene by combining 2 to 5 randomly selected objects. To enhance realism, we used random environment maps sampled from Polyhaven (Poly Haven, 2025) to serve as the background of the scenes. Additionally, we added a ground plane with a high-quality texture beneath the objects, using Perlin noise to enhance the surface and add realistic variations. Finally, each object was given a random rotation to serve as an augmentation of the level of the object to train the pose estimation module. This entire process resulted in a dataset of 200k scenes, comprising a total of 8 million images for model training.

### 3.3.3 TRAINING

We directly apply L2 loss to rotation, translation, and size, with equal weighting for each term. To demonstrate the superiority of our framework, we first train our model only on the 3D Front datasets (Fu et al., 2021) for fair comparison. We mix the datasets curated by MIDI3D (Huang et al., 2024) and Instpifu (Liu et al., 2022). We align their render results according to room IDs, resulting in 20K scenes. We take 1K scenes as test sets and the rest as training sets. We train the model from scratch for 25K steps until it converged. To extend the generalization on open-set, we further mix our 200K open-set datasets into the indoor datasets, and take 1K scenes as open-set test sets. We train the model from scratch for 40K steps until the model converged.

## 4 EXPERIMENTS

### 4.1 SETTINGS

**Datasets.** We conduct experiments on both indoor and open-set datasets. Specifically, we randomly select 1K scenes with no overlap with the training set from 3D-front (Fu et al., 2021) as indoor test sets, and 1K scenes from our collected open-set data as open-set test sets. It is worth noting that our 3D-Front scenes contain significantly more occlusions compared to MIDI. To further evaluate the generalization, we conduct qualitative comparison on synthetic, in-the-wild, and multi-scale images.

**Baselines.** We compare our method with the state-of-art methods (Lin et al., 2025; Huang et al., 2024) on both indoor scenes and open-set datasets. Since CAST3D (Yao et al., 2025) has not released its code or dataset, we can only provide qualitative comparisons in Figure 7.

**Metrics.** Following existing scene generation methods (Yao et al., 2025; Huang et al., 2024; Lin et al., 2025), we use scene-level Chamfer Distance (CD-S), F-Score (F-Score-S), and IoU Bounding Box (IoU-B) to evaluate the quality of the whole scene. And we use object-level Chamfer Distance (CD-O) and F-Score (F-Score-O) to evaluate the quality of generated object geometry.

### 4.2 QUANTITATIVE RESULTS

We conducted a quantitative evaluation of our indoor scene dataset against the standard 3D-Front (Fu et al., 2021) dataset. Since there is no existing open-set 3D scene generation benchmark, we constructed our own datasets specifically for this purpose. As shown in Table 3, our method consistently outperforms existing baselines, achieving the highest quantitative metrics for both indoor and the more challenging open-set scene generation tasks. Remarkably, even without being trained on the open-set dataset we constructed, our approach still obtained the best quantitative results. This underscores the superior performance of our proposed framework and designed modules.

### 4.3 QUALITATIVE RESULTS

As shown in Figure 7, our method generates visually compelling scenes that are not only realistic but also rich in detail. Crucially, our model demonstrates a robust ability to handle severe occlusions in Figure(a)(b), accurately reasoning about the relative spatial relationships between objects and places objects in plausible poses in Figure(c)(d)(f). Besides, our model can also handle small objects without geometry degradation in Figure(e).

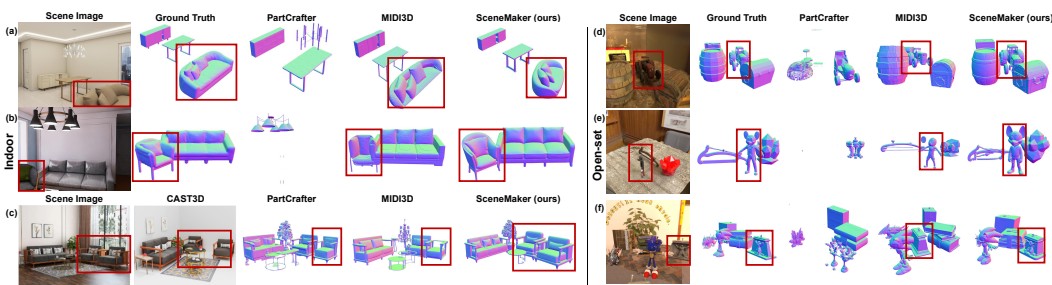

Figure 7: **Qualitative comparison with scene generation methods.**

## 4.4 ABLATION STUDY

**Attention Mechanism.** We ablate the contribution of the global and local self-attention mechanism, and the decoupled cross-attention mechanism in the pose estimation model respectively. For the self-attention mechanism, we simply remove the global and local attention modules respectively for comparison. For the

| Method | CD-S↓ | FS-S↑ | CD-O↓ | FS-O↑ | IoU-B↑ |
|---|---|---|---|---|---|
| SceneMaker | 0.0242 | 0.7502 | 0.0294 | 0.8121 | 0.7555 |
| w/o GSA | 0.0340 | 0.6610 | 0.0556 | 0.6293 | 0.7336 |
| w/o LSA | 0.0293 | 0.7434 | 0.0901 | 0.7142 | 0.7733 |
| w/o LCA | 0.0274 | 0.7368 | 0.0429 | 0.7113 | 0.7882 |
| + Complete points | **0.0064** | **0.9197** | **0.0124** | **0.8432** | **0.8550** |

Table 4: Quantitative results of ablation studies. **Bold** and underline indicate the best and the second best, respectively.

decoupled cross-attention mechanism, we remove the local attention and merge the rotation update into the global attention for comparison. We train the above models from scratch and use ground-truth meshes to eliminate the influence of geometry on pose estimation. As shown in Table 4, all modules in our proposed attention mechanisms contribute positively to model performance.

**Open-set Datasets.** We demonstrate the necessity of our proposed scene datasets on the open-set images as shown in Table 3. Our model faces severe degradation in open-set scenario without the datasets. The datasets mainly provide open-set patterns of diverse objects, which help build pose mappings across different geometries and are essential for open-set scene generation.

**Upper Bound of Pose Estimation.** Compared to a single image, videos or multi-image can provide richer scene structure information through point cloud reconstruction. When the reconstruction algorithm (Wang et al., 2025a; 2024) reaches its upper limit, it is equivalent to providing our model with a complete point cloud. We discuss the upper bound of our pose estimation model by giving the complete point clouds of the scene. As shown in Table 4, with a complete point cloud, our model achieves a significant performance boost, demonstrating its strong potential under video or multi-image conditions.

## 5 CONCLUSION

In this paper, we propose a decoupled 3D scene generation framework called **SceneMaker**. To obtain sufficient occlusion priors, we decouple and develop the robust de-occlusion model from 3D object generation by leveraging image generation models and a 10K curated de-occlusion dataset for training. To improve the accuracy of the pose estimation model, we propose a unified pose estimation diffusion model with both local and global attention mechanisms. We further construct a 200K synthesized scene dataset for open-set generalization. Comprehensive experiments demonstrate the superiority of our framework on both indoor and open-set scenes.

**Limitations.** Although our framework effectively generalizes to arbitrary objects, the real-world arrangement of objects is often much more complex than what our datasets capture, particularly when force interactions are involved. Therefore, a key future research topic is how to construct or refine 3D scenes more accurately in a physically plausible manner, including interpenetration and force interactions. Meanwhile, existing methods can only control scene generation through images or simple captions, and further development is needed for more control signals and natural language interactions. Moreover, how to perform more in-depth understanding tasks and adapt embodied decision-making based on generated high-quality 3D scenes is also an unsolved challenge.

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
