# OpenReview forum: "SceneMaker: Open-set 3D Scene Generation with Decoupled De-occlusion and Pose Estimation Model"
_ICLR.cc/2026/Conference — ICLR 2026 Conference Withdrawn Submission_

### Official Review · Reviewer_CA1e · 2025-10-29

**Soundness:** 3
**Presentation:** 2
**Contribution:** 2
**Rating:** 4
**Confidence:** 3

**Summary:**

This paper proposes a decoupled 3D scene generation framework. It tries to simultaneously produce high-quality geometry and accurate poses under severe occlusion and open-set settings. It divides 3D scene generation into de-occlusion, 3D object generation and pose estimation tasks, respectively. The authors first finetunes an image editing model to generate de-occluded object images, then adopts existing image-to-3d methods to generate object geometry based on the de-occluded object images, and proposes a pose estimation model to directly predict the object rotation, translation and size. Compared to the baselines, the proposed method can achieve better generation quality under occlusion.

**Strengths:**

1. By first generating de-occluded images and then doing 3D object generation, the quality and accuracy of individual 3D object geometry can be improved.

2. I appreciate that the authors curated datasets for both de-occlusion finetuning and pose estimation training, which also improve the performance.

**Weaknesses:**

1. From the teaser and the qualitative comparison, the visual quality of the proposed method improved over the baselines. However, the pose of the objects and the relations between the objects do not seem to be very faithful to the input scene image. For example, for the second image in the teaser, the orientation of the bookshelf seems to be not accurate; in the fourth image, the chair is overlapped with the table. Also, in Figure 7 (a) and (d) is not very easy to tell if the proposed method is better than the baselines.

2. There is no video / 3D that can show the scene quality from other viewing directions.

**Questions:**

I appreciate the idea of the decoupling the generation stages and the efforts of collecting respective datasets and finetuning the models. However, my main concern of the paper is that despite improving quality showcased by a few visual comparisons and quantitative numbers, I’m still not convinced that the quality of the proposed method is substantial enough. It makes me doubt that if the proposed methodology can really achieve a good and consistent improvement or not. Could the authors provide more qualitative examples, as well as the generated scenes from other viewing directions?

---

### Official Review · Reviewer_yS2A · 2025-10-31

**Soundness:** 2
**Presentation:** 2
**Contribution:** 2
**Rating:** 4
**Confidence:** 3

**Summary:**

The paper addresses the task of open-set 3D scene generation from a single image, i.e., generating a full 3D scene with arbitrary objects in  open-world domain. This paper proposes to decouple the scene generation pipeline into three stages: de-occlusion, 3D object generation, and pose estimation, each trained/finetuned on separate data sources providing the necessary supervisions. The authors develop a 2D de-occlusion model (to recover the full appearance of occluded objects) trained on large image datasets and a curated 10K de-occlusion dataset with diverse, severe occlusion patterns. Next, they leverage existing 3D generative models for object reconstruction from the de-occluded images. Finally, they introduce a unified pose estimation model that directly predicts each object’s 6DoF pose (rotation and translation) + size, combining both local object-specific and global scene-level attention to capture inter-object relationships. For open-set learning, they also create a large-scale synthetic scene dataset (200K scenes, 8 million images) by randomly composing objects with varied backgrounds, expanding the training data. Experiments show that SceneMaker achieves SOTA performance with higher-quality object geometry and more accurate object poses under heavy occlusions and in-the-wild open-set scenes, outperforming previous methods on standard indoor benchmarks and a new open-set test set.

**Strengths:**

- The idea of decoupling de-occlusion, 3D object generation, and pose estimation makes sense and is executed cleanly. The modular design lets each part specialize, and the authors make reasonable choices about what priors each module should learn from. In my opinion, the most meaningful contribution here is probably the data: the 10K de-occlusion dataset and especially the 200K synthetic scenes used for pose estimation. These fill a clear gap in prior work and seem to drive much of the model’s performance in open-set settings.

- Results are solid on both indoor and open-set scenes. The paper includes good ablations, e.g., removing global/local self-attention or the cross-attention structure in the pose module clearly hurts performance. Likewise, the gains from adding the synthetic scene data are clearly shown on the open-set test set. The qualitative results are also compelling. Compared to prior methods, the model does a better job of completing occluded objects. Small objects are handled better than in many earlier pipelines, which often miss or collapse them.

**Weaknesses:**

- Pipeline complexity: The proposed system involves many moving parts, eg, depth estimation, segmentation, a diffusion-based de-occlusion module, 3D object generation, and a custom module for pose estimation. While the modular design has it merits (see Strengths), coordinating all these components can make the pipeline hard to reproduce or extend. Each module requires either training or careful finetuning with custom training data, and errors in early stages (e.g., segmentation or depth estimation) could easily propagate downstream. The paper relies on strong pretrained models (e.g., Grounded-SAM, MoGe), but doesn’t evaluate how sensitive the system is to noise or failure in these early predictions. It would be helpful to understand whether the pipeline can recover gracefully, for example, if the de-occlusion step fails for an unusual object, does the geometry or pose prediction collapse, or does the system fail gracefully? Some kind of end-to-end refinement or joint training might help address this, but this isn't explored. It would be great to have some analysis on common failure modes.

- Sim2Real generalization: While the 200K synthetic scene dataset is a nice contribution, there are still questions about how well it captures real-world complexity. The synthetic scenes are constructed by randomly placing 2–5 objects on a plane with random rotations and lighting. This adds diversity, but lacks the structured layouts and rich interactions often seen in real environments, e.g., objects stacked or leaning against each other. The authors acknowledge this in the limitations section, noting that the synthetic data doesn’t model force interactions or complex arrangements. Because the pose model learns from scenes without inter-object interaction, it might struggle in real settings with more objects or unusual configurations. It’s also unclear how well the system would handle scenes with more than 5 or 6 objects as the training data doesn’t go beyond that. The paper shows good qualitative results on real images, but doesn’t test on truly cluttered or diverse real-world scenes, so generalization is a concern for me.

- Open-set training tradeoffs: One potentially concerning observation is in Table 3, where finetuning with the proposed open-set scene data appears to degrade performance on the 3D-Front benchmark. Therefore, it raises the question of whether the model is overfitting to the open-set data.

- Formatting/citation nitpicking: Flux kontext model from Black Forest Labs should **not** be cited as (*Labs et al., 2025*). Some other citations don't have the proper conference information: e.g., Midi is published in CVPR 2025 but is only cited as an arXiv preprint. Please do a thorough clean-up of the citation formatting.

**Questions:**

- Please see the details in the Weaknesses Section.

- Will the custom datasets and code be released? In my opinion, the 10K de-occlusion dataset and the 200K synthetic scene dataset for pose estimation are important contributions of the paper. Given their potential value to the community, it would be helpful to know whether the authors plan to release these datasets. Similarly, is there an intention to release code or pretrained models? Reproducibility could be a challenge given the system’s complexity, so open-sourcing would significantly strengthen the impact of the work.

---

### Official Review · Reviewer_wBht · 2025-11-02

**Soundness:** 1
**Presentation:** 3
**Contribution:** 1
**Rating:** 2
**Confidence:** 4

**Summary:**

This paper presents a three-stage pipeline for 3D object reconstruction that leverages several existing foundation models (Grounded-SAM for segmentation, DINOv2, and a point encoder) as features to condition a diffusion model for pose estimation. The final stage maps these results to a shared canonical space to predict the object's pose.The approach demonstrates effectiveness on the 3D-Front and a new dataset, achieving improvements compared to scene generation methods.

**Strengths:**

+ The idea is simple and writing is clear.
+ The paper achieves good numbers compared to the scene generation methods.

**Weaknesses:**

- **System Paper and Novelty**: The primary concern is that the work is a systems paper that assembles existing foundation models (Grounded-SAM, DINOv2, Diffusion, etc.) into a pipeline. The core methodology or models are not novel contributions by the authors. Simply constructing a de-occlusion dataset/test-set is likely insufficient for ICLR acceptance given this lack of foundational novelty. Suggestion: The authors should focus on improving or outperforming one of the constituent foundation models (e.g., Grounded-SAM, DINOv2, Diffusion) in their respective task to strengthen the contribution.

- **Evaluation Scope of Reconstruction**:
  The current reconstruction evaluation is restricted to indoor datasets (where depth maps are typically reliable).
  - It would be good to quantitatively evaluate on ARKITScenes to broaden the indoor test set evaluation.
  - Evaluate on outdoor datasets (depth $> 30$ m) where depth maps are significantly noisier, masks are occluded, and point clouds are less reliable. This is crucial for demonstrating robustness.

- **Pose Estimation Evaluation**:  Since the paper claims to have better pose estimation, how about using this method for Pose-estimation on the Omni3D [B] dataset and compare against the Cube R-CNN baseline. Note that Cube R-CNN ouputs 3D bounding boxes and so has 3D center, 3D pose and size of the object. The authors should compare the current method against Cube R-CNN [C] on the 3D pose and size estimation task on all six constituent datasets of the Omni3D dataset with a single model.

- **Missing Baselines**: The paper does not compare against notable related models such as the VGGT [A] / MapAnything models for 3D object reconstruction?

- **Overclaimed Decoupling**: L322 of the paper claims that using global and local attention "properly decouples different pose variables". It would be good if authors mathematically prove this decoupling and also, quantitatively evaluate the effect of this "proper pose decoupling."

References:
- A. Wang et al, VGGT, CVPR25.
- B. Brazil et al, Omni3D, CVPR23.

**Questions:**

Please see the weakness.

---

### Official Review · Reviewer_A4SB · 2025-11-03

**Soundness:** 2
**Presentation:** 3
**Contribution:** 2
**Rating:** 2
**Confidence:** 5

**Summary:**

This paper presents a framework named SceneMaker for open-set scene component reconstruction from RGB image(s). To address the challenging task, SceneMaker decomposes the pipeline into de-occlusion, 3D component generation, and pose estimation. For the de-occlusion stage, this paper leverages 2D image generation priors with specifically designed fine-tuning dataset. For pose estimation, a unified multi-modal pose estimation network is proposed. The quantitative results demonstrate SOTA performance.

**Strengths:**

1. The divide-and-conquer design, which aims to maximize the utilization of existing priors, is well-motivated, and the quantitative evaluation further validates its effectiveness.

2. The specifically designed dataset for de-occlusion and pose estimation could facilitate future research in related fields.

3. The unified pose estimation network achieves strong performance, which may inspire future researches on scene-level pose estimation.

**Weaknesses:**

1. In L85, the author claims that SceneMaker is the first decoupled framework to divide the task into de-occlusion, 3D generation, and pose estimation, which is inaccurate. Early work such as Gen3DSR[1] also adopts a divide-and-conquer strategy.

2.In L154-155 and L202-204, the authors state that CAST3D[2] lacks interaction between objects, which is misleading. Section 5 of the CAST3D[2] paper explicitly describes post-processing procedures to adjust the layout under physical constraints.

3.There are no sufficient details about the de-occlusion model design, therefore, it is hard to justify the contribution. Also, the quantitative evaluation in Table 2 is confusing, as MIDI is a scene-level generation method and Amodal3R is object-level, while 3D-Front is a scene-level dataset. It is unclear how de-occlusion generation is appropriately evaluated on this dataset.

4.There is no description about the 3D generation model used in the overall pipeline.

5.In Table 3, the metrics for MIDI[3] are much lower than those reported in the original MIDI paper. Although the authors argue that the test set contains more occlusions than that from MIDI paper, it would be more convincing to also provide results on the original test set from MIDI.

6.Following point 1, there is no evaluation about Gen3DSR[1], leading to insufficient comparison. Additionally, in L412-413, the authors mention qualitative comparisons with CAST3D[2] in Figure 7, but these are missing.

7.Qualitative results in Figure 1 and Figure 7 also raise my concerns. In Figure 1 (last row, fourth column), there exists geometry interpenetration between the chair and the desk, indicating a failure case of the unified pose estimation. Moreover, in Figure 7 (e) and (f), the generated instances differ significantly from the ground truth, suggesting limitations in the generation module.
[1]: Ardelean, A., Özer, M., & Egger, B. (2025, March). Gen3dsr: Generalizable 3d scene reconstruction via divide and conquer from a single view. In 2025 International Conference on 3D Vision (3DV) (pp. 616-626). IEEE.
[2]: Yao, K., Zhang, L., Yan, X., Zeng, Y., Zhang, Q., Xu, L., ... & Yu, J. (2025). Cast: Component-aligned 3d scene reconstruction from an rgb image. ACM Transactions on Graphics (TOG), 44(4), 1-19.
[3]: Huang, Z., Guo, Y. C., An, X., Yang, Y., Li, Y., Zou, Z. X., ... & Sheng, L. (2025). Midi: Multi-instance diffusion for single image to 3d scene generation. In Proceedings of the Computer Vision and Pattern Recognition Conference (pp. 23646-23657).

**Questions:**

1. Please revise the misleading descriptions throughout the paper.

2. Please provide detailed explanations of the de-occlusion model and the 3D generation model used.

3. Please include the missing evaluations: comparisons with Gen3DSR, results on the original MIDI test set, and qualitative comparisons with CAST3D.

4. Please address the concerns raised in Weakness 7.

5. Please add some exhibitions about the de-occlusion fine-tuning dataset and scene dataset for pose estimation.

6. How many objects can the pose estimation model handle in one forward pass? Since the model is trained on datasets containing only 2–5 objects, it raises concerns about its scalability and performance on real-world scenes, which typically include far more objects.

---

### Note · Authors · 2025-11-12

I have read and agree with the venue's withdrawal policy on behalf of myself and my co-authors.